# Bat teeth illuminate the diversification of mammalian tooth classes

Alexa Sadier [1] ✉, Neal Anthwal [1,2], Andrew L. Krause[3], Renaud Dessalles[1,4], Michael Lake[5], Laurent A. Bentolila [5], Robert Haase [6], Natalie A. Nieves[1], Sharlene E. Santana [7] & Karen E. Sears [1] ✉

Tooth classes are an innovation that has contributed to the evolutionary success of mammals. However, our understanding of the mechanisms by which tooth classes diversified remain limited. We use the evolutionary radiation of noctilionoid bats to show how the tooth developmental program evolved during the adaptation to new diet types. Combining morphological, developmental and mathematical modeling approaches, we demonstrate that tooth classes develop through independent developmental cascades that deviate from classical models. We show that the diversification of tooth number and size is driven by jaw growth rate modulation, explaining the rapid gain/loss of teeth in this clade. Finally, we mathematically model the successive appearance of tooth buds, supporting the hypothesis that growth acts as a key driver of the evolution of tooth number and size. Our work reveal how growth, by tinkering with reaction/diffusion processes, drives the diversification of tooth classes and other repeated structure during adaptive radiations.

From the conical shape of the earliest vertebrate teeth, mammals have evolved a heterodont dentition with four tooth classes (incisors, canines, premolars, and molars), each with distinct morphologies that allow for specific functions during food processing[1]. This innovation enabled mammals' evolution of complex teeth with a wide diversity of morphologies and their subsequent utilization of a broad range of dietary sources; it is therefore considered a key innovation in the evolutionary success of the group[1–3]. In the last 30 years, the study of vertebrate tooth development has led to new insights regarding the evolution of teeth in various clades[2,4–6]. However, our understanding of the origin and diversification of mammalian tooth classes remains limited in large part because most developmental studies on mammalian teeth have focused on mice. With their derived, reduced dentition containing only molars and extremely modified ever-growing incisors, mice make a less than ideal model system for studying the origins of mammalian tooth classes. The field of evo-devo therefore needs a mammalian model with a complete dentition with which to

study the developmental foundation of the evolution and diversification of tooth classes.

The ideal mammalian group to fill this gap would possess a complete dentition (e.g., all four tooth classes), a large morphological variation in this dentition, and accessible development from a morphological and experimental point of view. Noctilionoid bats meet all of these requirements. Emerging 45 million years ago, noctilionoid bats underwent a major evolutionary radiation such that today their more than 200 species utilize nearly all possible mammalian diets (i.e., fruit, nectar and pollen, leaves, seeds, arthropods, small vertebrates, fish, and even blood)[7] (Fig. 1 and Supplementary Figs. 1 and 2). Similar to the evolution of disparate, diet-related beak shapes in Darwin's finches[8], noctilionoid bats have evolved a wide diversity of skull shapes to meet their dietary needs[7] along with a rich diversity in the proportion, size, shape, and number of teeth, particularly those used for mastication (premolars and molars; Fig. 1 and Supplementary Figs. 1 and 2)[9–12]. In association with this variation in tooth number and size,

[1]Department of Ecology and Evolutionary Biology, University of California Los Angeles, Los Angeles, CA, USA. [2]Centre for Craniofacial and Regenerative Biology, King's College London, London, UK. [3]Mathematical Institute, University of Oxford, Oxford, UK. [4]Greenshield, 46 rue Saint-Antoine, 75004 Paris, France. [5]Advanced Light Microscopy and Spectroscopy Laboratory, California NanoSystems Institute, UCLA, Los Angeles, CA 90095, USA. [6]DFG Cluster of Excellence "Physics of Life", TU Dresden, Dresden, Germany. [7]Department of Biology and Burke Museum of Natural History and Culture, University of Washington, Seattle, WA, USA. ✉e-mail: asadier@ucla.edu; ksears@ucla.edu

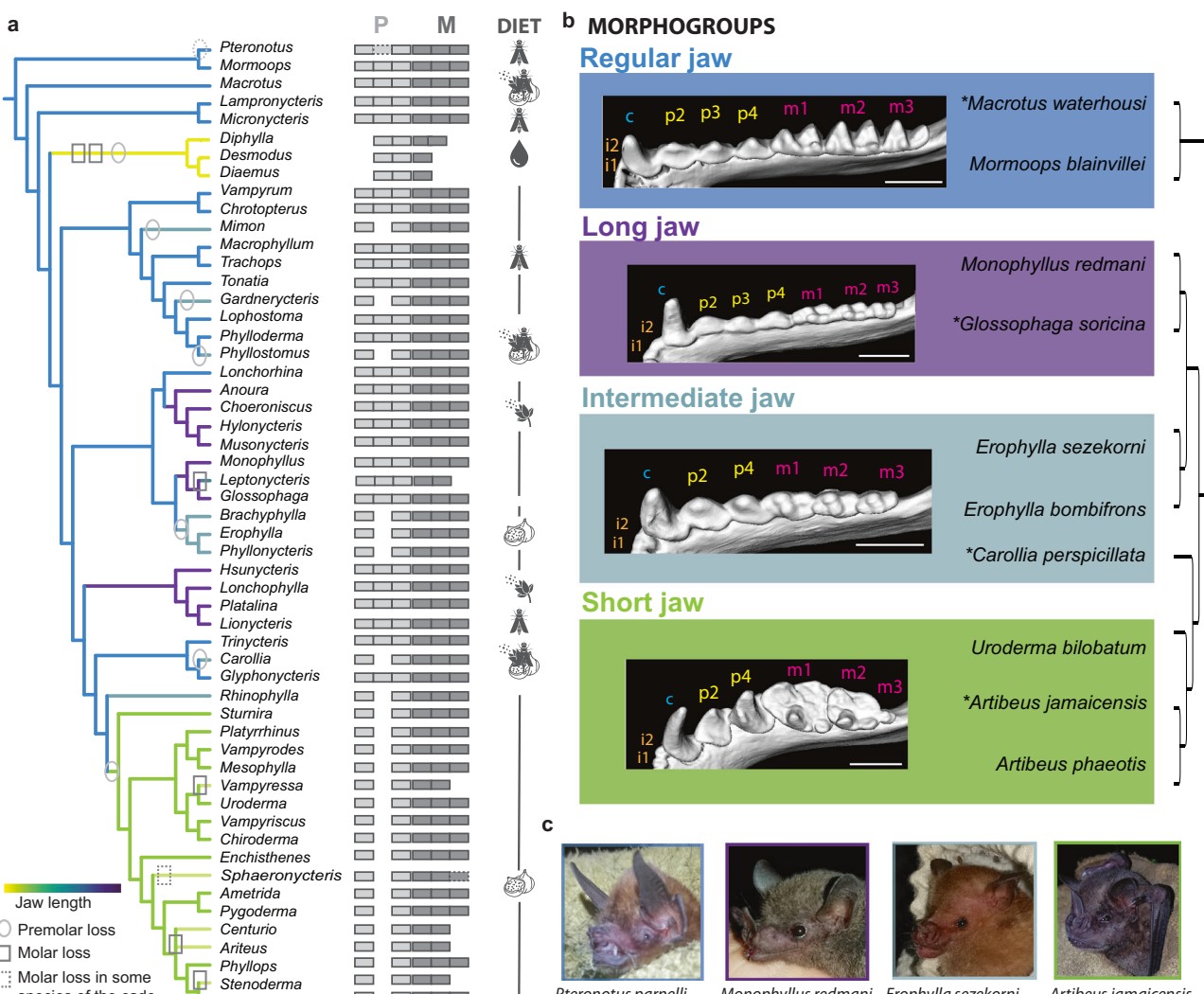

**Fig. 1 | Dentition diversity in noctilionoid bats. a** Tooth formula and jaw length of noctilionoid bats. Jaw length is represented with a color code (yellow to purple, short to long) and premolar and molars losses events by circles or squares respectively. Some losses happened independently in different clades. Diet is indicated with icons. **b** Morphogroups used in this study: regular jaw (3P3MR), long jaw (3P3ML), intermediate jaw (2P3MR) and short jaw (2P3MS and 2P2MS). Tooth classes are indicated by a letter: i, incisor; c, canine; p, premolar; m, molar. Representative genera and species investigated during development. Scale bar: 2 mm. **c** Picture of four species representative of the four different morphogroups that exhibit differences in jaw size and tooth composition. Source data are provided as Source Data File 1. Icons: Fig (CC BY 3.0), created by Linseed Studio from Noun Project; Pollen: (CC BY 3.0), created by Lars Meiertoberens; Blood (CC BY 3.0), created by romzicon.Source Data File 1.

rostrum and jaw length are highly diverse[13], and range from extremely elongated in nectarivorous species, to highly shortened in durophagous frugivores[14]. This last aspect of cranial morphological diversity in particular has been associated with variation in cell division rates, linked to variation in heterochronies during development[11,15,16]. Finally, bats have kept a deciduous "milk" dentition at several locus, in particular for two of the premolars, a feature that is ancestral to mammals[17]. Together these traits make noctilionoid bats an ideal model with which to study the diversification of the patterning of mammalian tooth classes during evolutionary radiations.

As with other ectodermal appendages, teeth are patterned through reaction/diffusion or Turing[18] mechanisms that have been shown to regulate the induction, spacing and proportions of organ signaling centers[19–30]. In particular, the investigation of molar development in mice, has revealed that molars are patterned through a simple reaction/diffusion (Turing pattern) based rule, called the developmental inhibitory cascade (IC), which controls the successive emergence of molar signaling centers as the dental lamina grows and ultimately the respective molar proportions[29]. This rule has been tested and predicted the size of molars in various mammalian clades suggesting that this particular cascade type might be the most common patterning mechanism of molar development, as well as potentially other tooth classes[31,32]. However, further studies in other mammalian clades have revealed instances in which molars explore areas of morphospace beyond those predicted by the cascade, suggesting that the cascade itself is variable between groups and/or interacts with other developmental mechanisms to produce the tremendous variation of tooth morphologies seen in nature[33–36]. Regarding the patterning of other tooth types, the situation remains unclear. While the IC has been able to predict the size of premolars and molars in hominins (based on morphological measurements)[37], there is currently no developmental evidence revealing that this rule or a similar one (i.e., directional Turing-based directional cascades) patterns the successive emergence of other primary tooth signaling center beyond molars. Together, these observations highlight a need to examine the developmental cascades regulating tooth patterning in diverse clades to understand the developmental foundations of tooth class patterning and diversification.

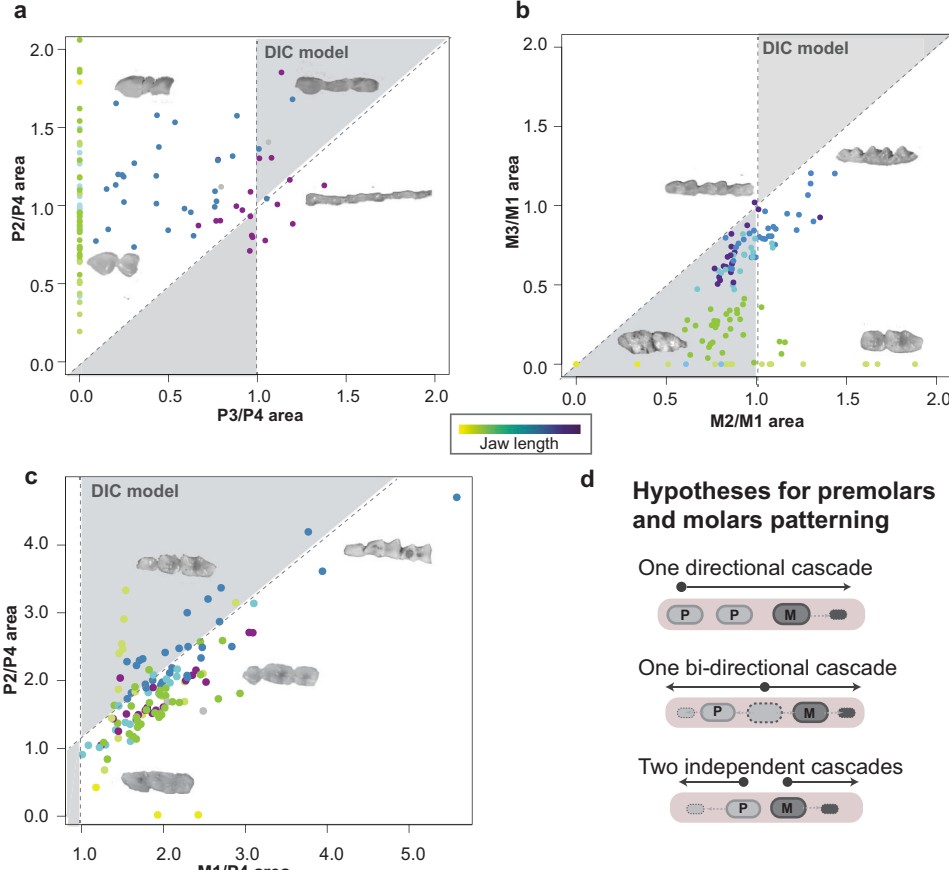

**Fig. 2 | Bat premolars and molars do not follow the classical IC model.** Testing the inhibitory cascade model on premolars (**a**), molars (**b**), and the P4-M1-M2 (**c**) triplet. Ratios of the T(n-1)/T(n) as a function of T(n)/T(n + 1) are represented. Species that follow the IC model colonize the gray triangles. **d** Hypotheses regarding the number of cascade-like Turing mechanisms that control premolar and molar development. Based on the research done in mice, premolars are molars have been hypothesized to develop through one unique inhibitory cascade that is initiated at the first premolar, here P2. The first scenario reflects this hypothesis with a unique cascade for premolar and molars initiated at the P2 antero-posteriorly. Alternatively, premolars and molar could develop through one initiator tooth (either the dP4 or the M1) in both directions (second scenario). Finally, premolars and molar could develop through two independent cascades in opposite directions initiated from two independent signaling centers, here dP4 and M1 (third scenario). Colors: jaw length as in Fig. 1. CT scans are available upon request.

In this study, we use the variation of noctilionoid bat teeth as a natural experiment to explore how the developmental cascades that pattern mammalian teeth and serial structures more generally have evolved during the diversification of tooth classes and adaptive morphological variation. We find that noctilionoid premolars and molars are patterned by distinct signaling cascades and that the distribution of tooth number and size differs in the premolars and molars of extant noctilionoids. In addition, by investigating the developmental processes driving these differences, we find results consistent with the hypothesis that, by perturbing the underlying Turing processes, growth modulates the number and size of the different classes of the teeth of noctilionoid bats during their adaptation to various dietary niches. Finally, we propose that such patterning could be modulated by evolution to produce morphological variation in other ectodermal appendages in bats and other mammals, as previously suggested but never demonstrated at a large taxonomic scale and in an ecological context[20,38,39].

## Results

### Noctilionoid premolars and molars deviate from expectations of the inhibitory cascade model and develop through two independent cascades

The work established by Kavanagh and colleagues has revealed that the co-variation of successive tooth proportions can be used as a starting point to infer the mechanisms by which teeth are patterned[29]. We thus decided to test for the existence of cascade-like co-variation patterns of tooth proportion in premolars and molars of bats. To test if two tooth classes, premolars and molars, develop through the same or distinct developmental inhibitory cascade(s) (IC), we calculated tooth area ratio (tooth length multiplied by tooth width) as done in refs. 29,37 for successive tooth triplets in the jaw: (i) the P2-P3-P4 or "premolar" triplet that (Fig. 2a), (ii) the M1-M2-M3 or "molar" triplet (Fig. 2b) and (iii) the P4-M1-M2 or "premolar-molar" triplet (Fig. 2c). Three models are possible to explain the development of post-canine teeth: (1) one initiation from the canine (as hypothesized based on mice whose development is very derived[40]) with a proximo-distal emergence of the buds; (2) one initiation that develops in both directions from dP4 or M1; and (3) two independent initiations, from dP4 and M1, that develop in opposite directions. If premolars and molars are initiated as in (1), we expect the triplets to follow the same or a very similar cascade. If premolars and molars are initiated as in (2) at M1, we expect the M1-M2-M3 (proximo-distal direction) or the P3-P4-M1 (disto-proximal direction) proportions to be linked. In both of these cases, we would also expect the proportions of the triplets to change in a linear manner following the relationship T(n + 2)/T(n)=2(T(n + 1)/T(n) −1, with T being the tooth and n being the first premolar or molar to develop[29] (Fig. 2d). If the two cascades are independent as in (3), we expect the cascades to follow different parameters. By measuring tooth areas, we found that variation in the adult premolar and molar proportions variation is not

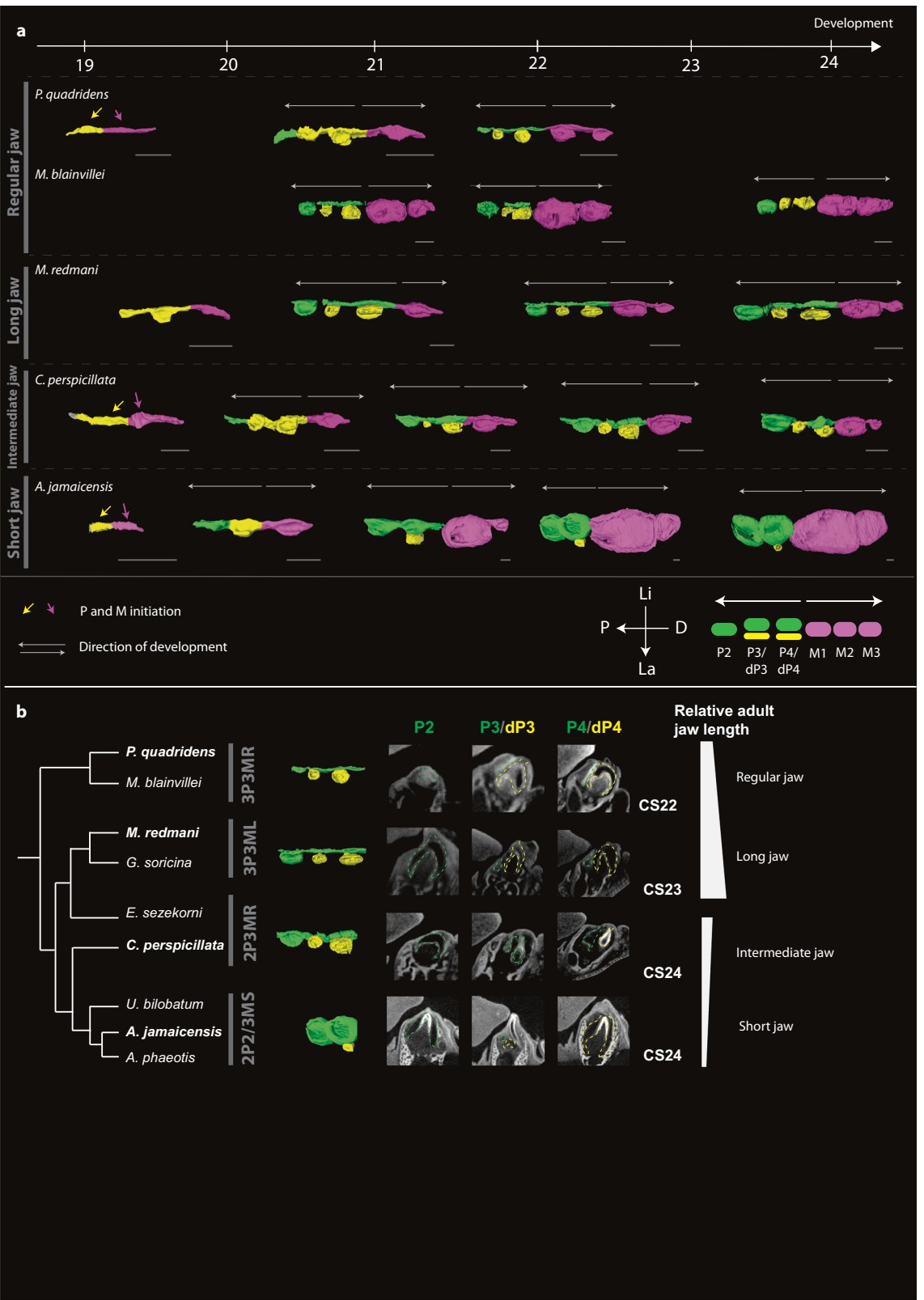

**Fig. 3 | Premolars and molars develop through two different cascades and are lost differentially as jaw length decreases. a** Reconstruction of the developing dental lamina in our four morphogroups of bats on µCT scans. Permanent premolars are indicated in green, deciduous premolars in yellow and molars in pink. Premolars and molars develop in two different directions, as the dental lamina grows and matures. Scale bar: 500 µm. **b** Gradual loss of dP3/P3 relative to adult jaw length. µCT slides of the different premolars are represented around CS23 for the different groups. The relative jaw length of adult bats is represented on the right, illustrating the relationship between tooth number during development and adult jaw length. Species represented in these µCT scan are in bold, others can be found in Supplementary Fig. 6. P posterior, A anterior, Li Lingual, La Labial. Each measure was taken three times. Source data are available in Source Data File 1 and 2. Detailed morphogroups are indicated in (**b**).Source Data File 1 and 2.

linear and that adult premolar and molar proportions occupy two different morphospaces (Fig. 2a, b and Supplementary Fig. 3). In addition, only 63.7% of species' molars and 8% of species' premolars fell within the expected IC morphospace (Fig. 2a, b). These findings suggest that the DIC rule is not sufficient to predict the morphological variation observed in the premolars and molars of noctilionoid bats and that the proportions of noctilionoid premolars and molars evolve independently. To confirm these findings developmentally, we used contrast-enhanced micro-computed tomography (μCT) scanning to study tooth development in eight species of noctilionoid bats that encompass much of the diversity in tooth number and size in the clade. We found that premolars and molars form from two distinct buds (Fig. 3a and Supplementary Fig. 4) that appear at the same developmental stage (Carnegie Stage 19, CS19) and seem to initiate two independent cascades (Fig. 3a) as the dental lamina grows in both directions (Fig. 3a). Together, these results suggest that the premolars and molars of noctilionoids are patterned by two different cascades that operate in opposite directions. This finding is consistent with molar and premolar patterning being largely independent from each other in noctilionoids, which is in contrast to what has been suggested in other mammals, for example, in hominins[37] or for the particular case of the lost premolar buds in mice[40]. Of note, P3 and P4 are replaced and exhibit a deciduous dentition[41–46]. As a result, the IC—if applicable—would likely directly influence the proportions of the first-generation of premolars (i.e., dP3 and dP4) and have potentially a distant effect on adult permanent premolars P3 and P4 which appear secondarily at the same locus.

## Premolar and molar numbers and proportions are correlated with jaw length in noctilionoids

As tooth proportions are controlled by the underlying developmental mechanisms that regulate their formation[29], we quantified morphological variation in the premolars and molars of 118 species ($N \leq 3$) of noctilionoid bats that span the ecological and dietary guilds found in the clade. Species were classified into four main morphogroups (see Supplementary regarding other morphogroups, Supplementary Figs. 1–3 and 5, Supplementary Table 1, Supplementary Data 1, 2, and 3) based on relative tooth number and jaw length (Fig. 1 and Supplementary Fig. 1): regular jaw, the ancestral pattern, composed of insectivores and omnivores with unspecialized jaw lengths compared to other guilds, three premolars and three molars; long jaw, consisting of nectarivores that have converged on an elongated jaw phenotype with three premolars and three molars; intermediate jaw, a convergent phenotype found in insectivores, omnivores, and frugivores with jaws of an intermediate length and two premolars and three molars; and short jaw, containing derived, frugivorous bats with short jaws and two premolars and two or three molars (the third one being reduced in size when present). We found the number of teeth to generally be associated with the size of the jaw among species (Supplementary Fig. 5); the jaws of 6-toothed long-jaw and regular jaw bats are significantly longer than those of 4- of 5-toothed intermediate jaws, and short jaw bats. Interestingly, some of these morphogroups, such as intermediate or short jaws, are polyphyletic and contain independent events of tooth loss, consistent with jaw length reduction having been being repeatedly associated with tooth loss in noctilionoids. In addition, we found that the proportions of individual premolars and molars are more disparate in species with shorter jaws (typically frugivores; Supplementary Fig. 3); bats with elongated jaws routinely have thinner and longer teeth with more similar proportions (long jaw, Supplementary Figs. 2 and 3 and Fig. 1). These findings are consistent with an association between the processes of jaw elongation and the generation of tooth proportion and number.

## Premolar and molar patterning and gain/loss is associated with dental lamina length

To explore the developmental mechanisms linking tooth number, tooth size, and jaw length, we investigated the possibility that the sequence of premolar and molar appearance during development depends on the available dental lamina space. As the dental lamina develops, the teeth appear successively so the space available at a given developmental time for the teeth depends on the growth rate of the dental lamina. In regular jaw bats, we observe that the development of dP4 is followed by the sequential development of dP3 and P2 (Fig. 3a, b) which is consistent with the predictions of the classic IC model and is that seen in mice molars, albeit in the reverse direction (i.e., from the back to the front of the jaw) (Fig. 2c). In intermediate jaw bats, which have slightly shorter jaws, further examination of the developing dental lamina reveals that, while the dP3 forms and mineralizes, its replacement adult P3 is initiated but does not develop further (Fig. 3b). At the extreme, in long-jaw bats, dP3 and P2 appear almost simultaneously since the jaw develops rapidly (Fig. 3a, b). Together, these results suggest that the development and timing of formation of different premolar buds is influenced by the space available in the jaw, likely because new teeth are able to form only at a certain distance from each other as predicted by the IC, and more largely, reaction/diffusion mechanisms that pattern ectodermal organs[29]. This finding is supported by the pattern of tooth formation in shorter-jawed bats; the development of the dP3 is initiated but the incipient tooth fails to grow and/or mineralize resulting in the loss of both dP3 and P3 (Fig. 3b and Supplementary Fig. 4). In some short jaw species, such as *Artibeus phaeotis*, the M3 is lost with no evidence that it ever started to develop. These patterns of loss are consistent with the variation observed in adult bats: the smallest teeth, specifically the middle premolar (P3) and the last molar (M3), exhibit the most variation in size among species with short jaws (Supplementary Fig. 3); P2 and P4 and M1 and M2, respectively, exhibit similar size variability among species with short and elongated jaws. Of note, the loss of the M3 is polymorphic in some species of short-faced bats, and this within-species variation has been linked to subtle variation in jaw size among individual bats[47,48]. These observations suggest that the presence/absence of M3 is dependent on the available space in the developing jaw (e.g., *Artibeus watsoni* appears to be just at the limit condition for which the M3 does or does not develop further). To sum up, premolars and molars exhibit divergent patterns of loss with decreasing jaw lengths, with premolars losing the middle tooth gradually (dP3 and then P3) and molars the last tooth of the row, the M3. These losses appear to have happened convergently in bats with similar jaw lengths.

## The number and proportion of premolars and molars is associated with variation in growth rate

The results presented above support the hypothesis that the numbers and proportions of premolars and molars are linked to the length of the jaw and, in particular, how fast the jaw is growing and where that growth is located along the jaw at the time of tooth bud formation. To further explore this idea, we measured the growth rate of the premolar area during development in 3D models reconstructed from phoso-photungstic acid-contrasted μCT scans in eight focal species representatives of our four morphogroups (see "Methods"). These measures reveal that species with jaws of average length (regular jaw and intermediate jaw) exhibit a moderate peak of jaw growth around stage 20 as dP3 and P2 develop (Fig. 4a). In long-jaw bats, this peak is three times faster than that of intermediate jaws and corresponds to the almost simultaneous formation of dP3 and P2. In contrast, no growth peak was observed in short-faced bats with a short jaw; the growth rate was lower in short-faced than other bats in this region. To examine these patterns in more detail, we used EdU and PCNA labeling to trace cell proliferation during jaw development (Fig. 4b, c and Supplementary Fig. 6). We found that long-jaw bats seem to exhibit higher rates of cell

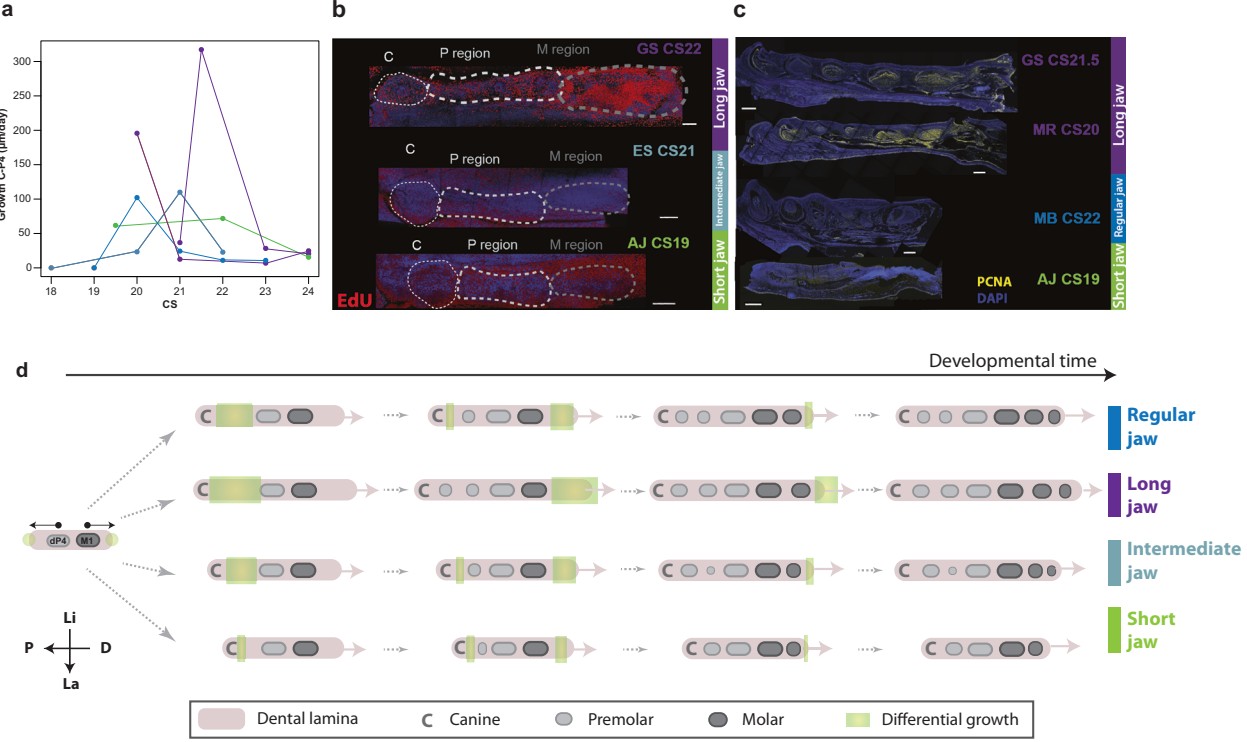

**Fig. 4 | Growth variation in the premolar region. a** Growth rate between the Canine (C) and the P4 position during development for species in our four groups calculated from the µCT scans. **b, c** Cell division patterns in the premolar and molar regions at relevant stages for *Glossophaga soricina* (GS, long jaw), *Monophyllus redmani* (MR, long jaw), *Mormoops blainvillei* (MB, regular jaw), *Erophylla sezekorni* (ES, 2P3MR) and *Artibeus. jamaicensis* (AJ, short jaw). To visualized cell division patterns during the development of the bats, pregnant females have been injected with EdU which is then visualized in whole mount dissected jaw (experiment performed three times with two replicates) (**b**). Alternatively, dividing cells were labeled using an antibody against PCNA in cryosections (experiment performed three times with two replicates) (**c**). The position of the canine is indicated with a white dotted line. CS: Carnegie Stage. Scale bar: 200 µm. **d** Model of the dynamics of emergence of premolar and molar buds and differential growth, based on µCT-scan measurements (**a**) and cell division experiments (**b, c**). Dental lamina develops in both directions from the initial dP4 and M1 tooth buds independently for premolars and molars. Differential growth rate for the different groups is indicated in green. Source data for (**a**) are available as Source Data file 3.Source Data file 3.

division in the premolar area compared to species with other morphotypes, while short jaw bats tend to show the lowest rates of cell division. These results echo previous results in long-jawed nectarivorous noctilionoids[15] and support the idea that differences in cell proliferation or growth rate contribute to and possibly drive craniofacial[16] and tooth size[32] differences between species (Fig. 4d).

### Growth rate likely tinkers with Turing patterns and thereby modulates the appearance order and size of teeth

Our findings suggest that the growth rate of the jaw might perturb the reaction/diffusion mechanisms that influence the sequence of tooth appearance and tooth proportions. As tissue growth rate have been shown to influence the activation/inhibition processes that pattern repeated structures during development in model species[20,27,38,49], we postulate that the variation in tooth number and size observed in noctilionoid bats could possibly be simply explained by growth rate-induced perturbations on the Turing mechanisms behind the ICs during the evolution of this clade. To begin to test this, we computationally examined if we could reproduce the various phenotypes observed during development by modeling variation in jaw size and growth. We implemented a simple model of a Turing-type reaction-diffusion system (Supplementary Data 1) to recapitulate the different sequences of insertions observed for the two independent cascades for premolars and molars (Fig. 5a, b). Using only apical growth, we successfully recapitulated two of the four morphogroups (corresponding to regular jaw *Pteronotus quadridens* and long-jaw *Monophyllus redmani*) (Fig. 5a). Presupposing exogenous spatial gradients (plausibly corresponding to a pre-pattern or additional spatial

modulation), we also captured the insertion sequence and—qualitatively—the size variations observed in short jaw *Artibeus jamaicensis* (Fig. 5b). These results demonstrate that, while there are undoubtedly other factors influencing the insertion and modulation of tooth signaling centers (which could explain the regular group with 3 premolars and 2 molars that loses a molar while having a long jaw with diastemas; Supplementary Figs. 2, 3, and 5), simple models combining reaction-diffusion processes and differential growth are sufficient to explain much of the observed variation in noctilionoid teeth. This supports the importance of growth rate and jaw length variation in modulating the number and size of both premolars and molars in two distinct ways (Fig. 6).

### Discussion

From a morphological point of view, the differences between tooth classes have often been assessed through studies of modularity[50–52] or through the inhibitory cascade model[37,50]. These methods commonly identify modules specific to the incisors and canines but fail to identify distinct premolar and molar modules, possibly because they lack the precision to distinguish these classes and/or the morphological and perhaps genetic differences between these latter tooth types are less pronounced than for other classes. On the other hand, developmental studies, largely based on morphological observations of the dental lamina of species[53] including the ferret[54], shrew[55,56], straw-colored fruit bat[17], opossum[57], and this study reveal that lower premolars and molars tend to develop in opposite directions, and thereby support the hypothesis that premolar and molar placodes are patterned through independent activation/inhibition mechanisms.

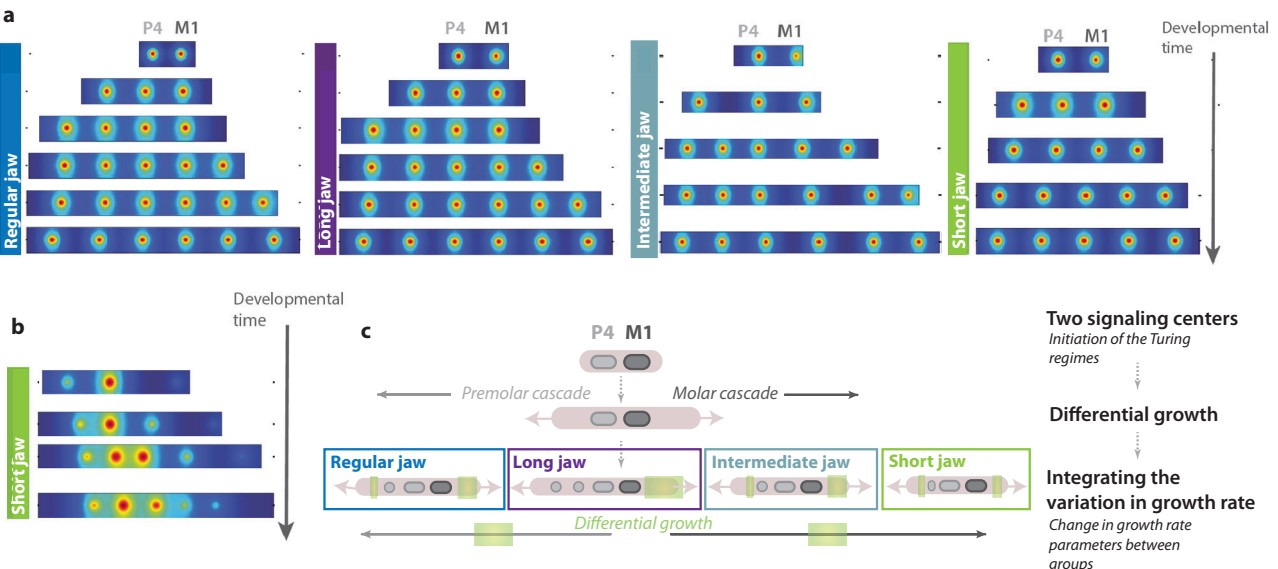

**Fig. 5 | Modeling growth rate in different species is sufficient to recapitulate the variation in tooth number and size observed in Noctilionid bats.**
**a** Reconstruction of the developmental sequences of the different morphogroups (regular jaw, long jaw, intermediate jaw, and short jaw) in our Turing-like model. Each simulation incorporates differential growth and is able to recapitulate the sequential appearance of premolars and molars in the different morphogroups. Inhibitory fields are represented by the circles. **b** Reconstruction of the sequential addition of tooth signaling centers of *Artibeus jamaicensis*, including tooth size

variation, using Turing patterns with differential growth. Inhibitory fields are represented by the circles. **c** Model: premolars and molars develop through two independent cascades, each of them having its own growing domain. Our model incorporates the variation in growth rate between the different species (growing domain is indicated in green) and is able to reproduce the diversity of tooth number and size seen in the different morphogroups. Modeling materials are available in Supplementary Note 1.

From a molecular perspective, some evidence suggests that the early pre-patterning of the jaw is driven by a "homeobox-code" that divides the jaw into territories that play a major role in tooth classes fate prior placode development (reviewed in refs. 58–60). While these results suggest that tooth class specification and/or determination occur prior placode induction, results of other studies suggest that this early code is not sufficient to establish tooth class identity and that other mechanisms, later during development, act in tandem to determine tooth class identity[60,61] Indeed, recent studies in other organisms such as the lizard genus *Pogona* have demonstrated that a heterodont dentition can be achieved through other mechanisms, such as a simple modulation of Eda signaling during later tooth development[62]. In opossum[57] and the lesser shrew[61], shifts in gene expression patterns have been observed between tooth classes, suggesting that different core developmental programs control their formation and that later steps of tooth development, such as morphogenesis are likely to play a role in tooth class determination. More recently, transcriptome analysis done in cats at the early bell stages have revealed differences in gene expression between tooth classes during morphogenesis[63]. Together, these results suggest that events that occur during later tooth formation and morphogenesis (e.g., bud, cap and bell stages) likely also play a major role in the establishment of tooth identity, in addition to earlier events such as the hox code. However, how the pre-patterning of the jaw and these later events are linked remain unknown[60] and should be the subject of future studies. The work we present here, which suggests that premolars and molars are patterned independently by two different cascades, supports the hypothesis that the initiation and morphogenesis of at least the premolar and molar tooth classes are largely independent and uncovers a mechanism that could act in tandem with the homeobox code and other processes to modulate that development.

Rapid diversification in the number and size of teeth is common during the colonization of different ecological niches and the associated incorporation of new food sources into the diet[1,64–66]. However, little is known about how this diversification could be facilitated by the

developmental mechanisms controlling tooth development. In particular, molar development and proportions have been shown to be constrained in mammals through the IC[29,31,32,67], making it traditionally difficult to explain the exploration of new areas of morphospaces. While the expansion of studies to include more clades has revealed that some seemingly do not follow the expectations of the IC model[33,35,36,67], the mechanisms by which these species escape this developmental bias remain unclear. Here, we provide some possible resolution to this conundrum by showing that growth, by perturbing the directional activation/diffusion mechanisms that pattern tooth development, could act as a simple modulator of tooth number and size and push tooth proportions into new areas of morphospace that are not predicted by the IC itself. In addition, we show that premolars and molars likely develop through two different activation/inhibition mechanisms or Turing-like cascades, with premolars being largely divergent. This finding helps explain observed, basic morphological differences between premolars and molars and their evolutionary changes (e.g., proportion, loss) coincident with the adoption of new food sources. Further studies will help to decipher the genetic control of these differences at early and late developmental stages.

The divergence observed for premolars could also be explained by the deciduous dentition, which has been poorly investigated in relation to the developmental cascade[37]. P3 and P4 exhibit deciduous teeth, respectively dP3 and dP4, and are thus not directly impacted by the initial cascade. As permanent premolars appear at the same locus as the deciduous dentition, our results predict accurately the sequence of apparition of the different premolar signaling centers. The rarity of juvenile specimens with erupted dP3 and dP4 make it difficult to study the cascade on the deciduous dentition as done in ref. 37. Moreover, bat deciduous dentition is heavily derived due to the constraints of flight: deciduous premolars have a "hook" shape that allow the pups to attach to the mother's nipple during flight[41–46], and which does not resemble the adult permanent dentition. As deciduous teeth and permanent teeth develop with closely conserved signals[2], we hypothesized that

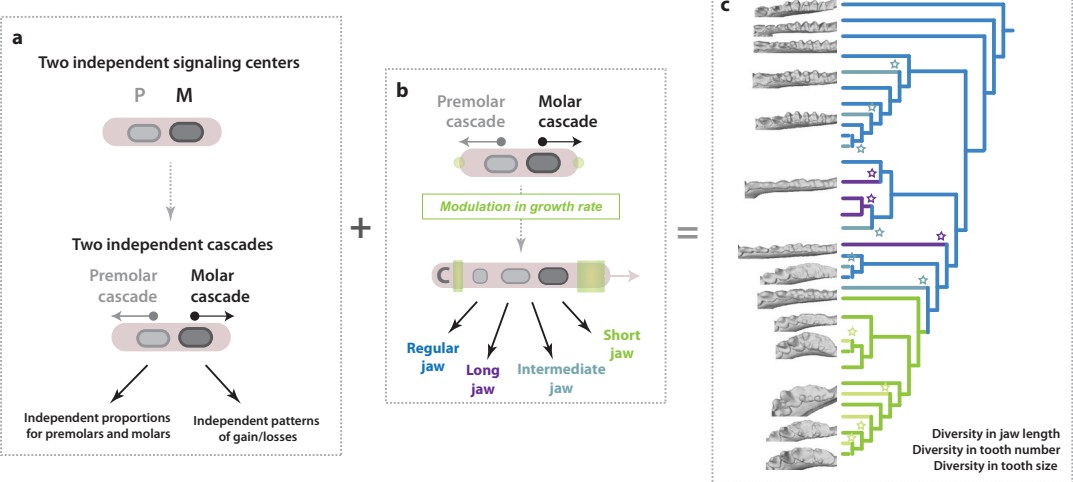

**Fig. 6 | Variation of developmental cascades explain the differences between premolars and molars and their evolution. a** Model for the patterning of pre-molars and molars. Premolars and molars develop through two independent cascades that each have their growing domain. **b** Variation in growth rate explain the phenotypes and pattern of tooth losses found in the different morphogroups in relationship with jaw length. **c** Variation of both the IC cascade and growth rate explain the diversity of premolar and molar number and size observed in Nocti-lionoid bats as well as how same morphologies have been reached convergently (stars).

the development of the permanent P3 and P4 signaling centers could still influence each other. Future studies will benefit from investigating how the patterning of permanent teeth is influenced by the deciduous dentition and/or if permanent teeth interact in a "Turing-like" manner.

In our study, we showed results consistent with the hypothesis that the rapid evolution and diversification of tooth number and size during the evolutionary radiation of noctilionoid bats occur through the modulation of Turing patterns, growth rate, and the resultant jaw space (Fig. 6). This finding is in accordance with the patterns of tooth appearance in the upper jaw of shrews[55,56]. In shrews, premolars appear antero-posteriorly (while premolars develop postero-anteriorly in the lower jaw) with P3 appearing before P4, suggesting that the first tooth from each class to appear could initiate its own cascade whose direction will depend on the space available in the jaw[5,56]. This could also explain how the direction of the addition of new teeth could change between species. Beyond teeth, this finding is consistent with what has been suggested for other ectodermal appendages (e.g., palatal rugae in mice and hamsters), in which a growth burst drives the appearance of new segments in the resulting available space[27,68], and in other ectodermal organs[23,49] that follow Turing pattern formation but without directional growth. As both the IC cascade and growth are implicated in the development of ectodermal appendages in general[24,29–31], our findings are potentially applicable to other ecto-dermal organs and propose a testable model to explain how their number and proportions can rapidly evolve, simply by modulating growth rate.

Our work demonstrates how new morphologies can potentially be rapidly achieved through subtle changes in the interaction between multiple developmental mechanisms during the bursts of diversifica-tion that often accompany evolutionary radiations (Fig. 6). While stu-dies of developmental rules often solely focus on the evolution of one developmental process to explain the evolution of characters, our work reveals the importance of studying the complex interaction of different developmental processes to fully understand the evolution of new morphologies during the colonization of new ecological niches, and identify bat teeth as a new model system to study these questions. Further work should focus on the identification of the molecular basis of these processes and how they interact with other mechanisms that likely play a role in morphological evolution.

## Methods
### Museum and field specimens
Museum specimen pictures have been: (1) taken from the FMNH in Chicago with a Nikon camera, (2) downloaded from Animal Diversity Web https://animaldiversity.org/ (see Supplementary Fig. 2). Field specimens were collected in the field (see Supplementary Data 1) using mist nets, harp traps or butterfly nets and euthanized humanely with isoflurane according to approved institutional animal care and use committee (IACUC) protocols 14199 at UIUC, 2017-093 at UCLA and the following permits Dominican Republic: VAPB-01436; Puerto Rico: 2015-EPE-028; Trinidad: 000619 and 000620 April 18, 2018. Speci-mens were then fixed ON at 4 °C with PFA and dehydrated the next day in 100% methanol and stored at −20 °C until used.

### Bat groups
We grouped species in six morphogroups representative of their dental diversity (Supplementary Fig. 2 and Supplementary Data 2) relative to tooth number and jaw size. Regular jaw represents the ancestral pattern of three premolars and three molars and a regular jaw length. Long jaw represents bats with three premolars and three molars with elongated jaws, as seen mostly in nectar feeders. Teeth are generally elongated and compressed laterally. Intermediate jaw represents bats with two premolars and three molars with a shorter jaw length, and short jaw. Short jaw represents bats with two premolars and two or three molars, with a short face and jaw. When present in short jaw bats is extremely reduced. The post-canine teeth are gen-erally wider.short jawshort jawIn our dataset, we also found two spe-cies (*Leptonycteris nivalis* and *Leptonycteris yerbabuenae*) with three premolars and two molars, 3P2M. In these species, the P3 is extremely reduced and their jaw size is closer to the second group. Because this morphology is marginal, it has not been used for the following analyses but was kept for the modeling. Finally, we excluded the vampire bats (*Desmodus rotundus*, *Diaemus youngi* and *Diphylla ecaudata*) from our analysis given the lack of embryos and their extremely reduced and derived dentition (with two premolars and one or two molars) that limit our developmental investigations. Diet groups were assessed based on ref. 14.

### Body mass
Body mass (Supplementary Data 2) was used to normalize the data in our analysis. Body mass data has been collected from Davalos and

Melo, Chapter 8[7], which gathered an impressive dataset on the body size of Noctilionoidea with the exception of: *Chilonycteris macleayii=Ptenoronus macleayii*[69]; *Leptonycteris nivalis*[70]; *Lonchophylla thomasii* (Emmons, 1990); *Platyrrhinus fusciventris*[71].

## Statistical analysis
Differences between the tooth areas of the different groups and teeth were compared by ANOVA and Tukey multiple comparisons of means (Supplementary Table 1 and Supplementary Data 3) in R.

## Developmental stages
Developmental stages for the different species of bats have been based on the development of *Carollia perspicillata* for which timed mating have allowed to assignment to the Carnegie Stage System[72]. These stages are relevant across mammals and for other species of bats[16,73]. Potential differences that could result from differences in developmental timing have been carefully assessed for each stage and each species on morphological and dental criteria's. All bats have been staged by visual inspection in the field (in addition to other metrics such as the size of the embryonic sac or the placenta) or on museum specimens (visual inspection only) using the various character observed in this staging system.

## μCT scanning and dental lamina segmentation
Bat embryo jaws were dissected and stained in 0.3% (phospho-tungstenic acid, Sigma) PTA in 70% ethanol (museum specimens) or 100% methanol (field samples) for 24 to 36 h on a rocker at room temperature. Stained specimens were mounted in a 1.5 or 2-mL eppendorf tube between two pieces of foam and μCT scanned in a Skyscan 1172, a Scanco uct50, or a Xradia BioMicroCT. Scan parameters were adjusted depending on specimen size and morphology, and voxel size ranged from 1 to 5 μm per scan. Raw μCT-scan shadow images were reconstructed to slices in NRecon, then imported into Mimics, where the dental lamina was segmented using the lasso tool every 4 to 5 slides before using the interpolation tool. Surface (stl) files were exported and used for visualization and morphological comparisons.

## Dental measurements
Adult dental measurements were taken from museum specimen photos (see Supplementary Data 1 for the specimens list) or Animal Diversity Web (animaldiversity.org) photo using ImageJ. Scale was set using the scale bar on the pictures. Crown width and length were measured three times for each tooth to ensure reproducibility. For analysis, individuals from the same species and locality were aggregated. Jaw length was measured from the tip of the jaw to the middle point between the left and right angle of the jaw. Tooth area was calculated by multiplying the length by the width of each tooth.

Embryo dental measurements were taken from reconstructed.stl files in mimics using the measurement tool. Each distance was measured three times, to ensure reproducibility, between the primary enamel knots or its resulting cusp to measure the distance between teeth, or the tooth's primary enamel knot and the end of the dental lamina to measure tooth sizes

Dental measurements are available in Source Data 1.

## HCR-IHC imaging
Field-sampled embryos were embedded in OCT and sectioned using a cryostat CM1520. Proliferated cells were detected using a PCNA antibody (Rabbit mAB #13110, Cell Signaling Technology) at 1:300, and the signal was amplified using the HCR system from Molecular Instrument according to the manufacturer protocol (using Donkey Anti-Rabbit Ab-B2)[74]. Sections were imaged using a Leica SP8 confocal.

## EdU staining
Pregnant females were injected in the field with 20 mg/kg of EdU reconstituted in DMSO and PBS in an intraperitoneal injection according to IUIAC procedures. Forty-five minutes after the injection, bats were euthanized with isoflurane. Embryos were dissected out of the female, and the jaws were carefully dissected and fixed overnight in 4% PFA at 4 °C before being dehydrated the next day in 100% methanol and stored at −20 °C until imaging. In the lab, half-dissected jaws were rehydrated and clarified using Scale S[75]. After the incubation in the S3 reagent, jaws were rehydrated in PBS before labeling. The next day was stained with a Click-iT EdU Alexa fluor 647 labeling kit according to the manufacturer protocol except for the incubation time, adjusted to 3 h, at RT. After the reaction, jaws were put in Scale S4 for the final clarification and imaging at the Leica SP8 DIVE two-photon microscope at the CNSI facility at UCLA using 760 nm and 1240 nm wavelength for Hoescht and Alexa Fluor 647, respectively. The resulting photos were processed using LASX and ImageJ.

## Modeling
See Supplementary information, Supplementary Note 1, and https://doi.org/10.5281/zenodo.8058070.

## Reporting summary
Further information on research design is available in the Nature Portfolio Reporting Summary linked to this article.

## Data availability
The μCT data are available under restricted access as other parts of the dentition/jaw are currently investigated by the authors, access can be obtained upon request to the corresponding author. The dental measurements data generated in this study are provided in Source Data file 1 and were measured on museum specimens or images from Animal Diversity Web (animaldiversity.org). Measurements on the μCT during development are provided in the Source Data file 3. Data files used for the simulation in Fig. 5 and Supplementary Note 1 are available on GitHub: https://doi.org/10.5281/zenodo.8058070. Source data are provided with this paper.

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

## Acknowledgements

We thank A. Couzens, A. Tucker, Y. Gibert, and V. Laudet for their critical reading of the manuscript. We thank L. Yin from X-ray imaging facility at UIUC, Beckman Institute for his assistance regarding CT scanning. Two-photon excitation laser scanning microscopy was performed at the Advanced Light Microscopy/Spectroscopy Laboratory (RRID: SCR_022789) and the Leica Microsystems Center of Excellence at the California NanoSystems Institute at UCLA with funding support from NIH Shared Instrumentation Grant S10OD025017. We thank N. Simmons for her helpful suggestions, museum specimen access and for organizing the field trip in Belize with B. Felton. In addition, we thank L. Rostant in Trinidad, M. Santiago and J. Almonthe in Dominican Republic, and A. Rodrigues in Puerto Rico for their support in the acquisition of permits for sampling and sample exportation. We thank N. Rochette for his help with statistical analysis and M. Oliva for her comments on the results. We thank the National Science Foundation (NSF) for supporting A. Sadier and K. Sears with grant award 201780 and S. Santana with grant award 2017738.

## Author contributions

A.S. and K.S. conceived the study. A.S. performed the investigations, designed and performed all the experiments and statistical analyses. A.S., N.N., S.S., and K.S. collected the measurement data. A.S. collected all the embryos used during fieldwork. N.A. provided additional specimens of C. Perspicillata. A.K. and R.D. coded and performed the modeling simulations. A.S., M.L. (core facility), and L.B. (core facility) performed the imaging. A.S. and R.H. processed the images. A.S. wrote the original draft and A.S., K.S., S.S., and N.A. reviewed and edited the manuscript. A.S. and K.S. administrated the project and acquired funding.

## Competing interests

The authors declare no competing interests.
