## [Peer Review File · Nature Communications]

Bat teeth illuminate the diversification of mammalian tooth classesREVIEWER COMMENTS

Reviewer #1 (Remarks to the Author):

The Sadier et al is a very interesting work that explores the diversity of bat dentitions, with a special focus on developmental mechanisms that might regulate and bias their diversity. Bats (here noctilionoids) show remarkable range of dental patterns, and although the phylogenetic and ecological factors underlying this diversity has been studied, the development of the dentitions has achieved much less attentions. This is undoubtedly much to do with the difficulty in obtaining material to study the process of bat tooth development. Here Sadier and co-workers have carried out a major effort to rectify the lack of data on the developmental basis of bat dental diversity. Just providing this basic data is a very valuable contribution worthy of publication. Analytically, the authors frame their results in the context of inhibitory cascade rule of sequential tooth development. They appear to argue that growth of the jaw is a major driver in the regulation of premolar and molar proportions. Below I have a few suggestions to clarify the text and the analyses.

1) Assuming that the premolar proportions follow the inhibitory cascade (IC).

IC mode of development assumes sequential development of individualized units with directionally biased effects. This mode of development is rather self evident for the molar teeth which bud off distally. In contrast, in premolars the cascading effects can be assumed to affect principally the deciduous premolars. Hence, why Evans et al 2016 examined deciduous premolars in hominins. Because permanent premolars develop as offshoots from their deciduous predecessors, the IC effects would be very indirect (at best), and I am not sure if there has been an explicit suggestion of permanent premolars following IC (at least I do not remember making one). That said, the analysis provided by the authors is totally valid, as they clearly demonstrate how premolars do not follow the IC (Fig. 2A). What might be in order is to slightly rephrase the text to state that permanent premolar development is not strictly a cascading process similar to molars. Thus, the primary hypothesis for this part of the analysis is not the IC, confirmed by their analyses.

2) Related to the point above, it is currently not clear in the text whether the deciduous dentition of the bats studied are functional. I recall that bat deciduous teeth tend to be reduced in number and size, and the beautiful and informative Fig. 3 seems to imply that this may be the case for at least some of the taxa studied. This would be valuable to point out. Also, if there are data to measure deciduous tooth sizes, this could be used to test the IC on those teeth. But I understand this data may be not available.

3) The Gierer-Meinhardt modeling is very well explained in the supplement. One detail that should be corrected in the main text (Page 2-3) is that the original IC model is not a dynamic reaction/diffusion model, but rather a rule assuming a simple activator/inhibitor balance in determining tooth proportions (this does not exclude the possibility of a Turing like process). Considering the modeling in the Sadier et al, the domains signifying tooth loci presumably denote to the placodes (initiation knots), although they could be considered to represent forming primary enamel knots too. In either case, the locations ('cells') representing signaling centers appear to move (change locations) in some of the model runs (e.g. SI Fig. 9). This is to be expected in dynamic systems without terminal differentiation. If this were the case in real jaws, the implication would be that the identity of placodal cells would shift before becoming fixed. Currently I did not notice mentioning of the dynamic nature of the modeling, and I leave it to the authors to decide whether something to point out.

4) Perhaps the most challenging part of the paper for a more general readership is the Morphogroups presented (3P3MR, 3P3ML...). These are very difficult to follow and the codes themselves are rather cryptic. It would make the text more readable if at least the main groups would be described in more understandable terms (superficially the dentitions seem to differ how laterally compressed they are). Also, currently the point colors in Fig. 2 seem to refer to the Morphogroups, although this is not stated. Interestingly, some groups (3P3ML?) seem to follow the IC prediction quite closely in Fig. 2B, as also many of the green group taxa where the M3 is highly reduced are close to the IC line.

5) There are some minor issues in text that could benefit from rewording. The use of superlatives such as astonishing, tremendous, groundbreaking, and outstanding is at least to this referee a bit distracting. Additionally, "key event" might be better to be "key innovation" (Introduction, first paragraph).

Reviewer #2 (Remarks to the Author):

Overall this is an intriguing paper. The authors have demonstrated an independent cascade pattern for the premolar region relative to the molar region in noctilionid bats, and developed a model based on reaction diffusion gradients in which jaw elongation can modulate the appearance of tooth loci. The data analysis and methodology seem to be sound and original. However, in my opinion, the conclusions are somewhat oversold as explaining more about the evolution of heterodonty and patterning in these regions. The authors admit near the end of the article, this is a complex problem and a number of factors are likely to operate, yet the bulk of the MS states that this model solves everything and we can ignore homeobox patterning completely, even though the model is operating much later in development than homeobox patterning. Further, it seems to take as given that the ICM is the ancestral pattern in mammals, while ignoring that it doesn't apply to many groups, and this warrants further discussion in the MS -- especially since their model only explains the patterns in 2 of the 4 morphogroups they establish, so only about half of the observed range of variation.

In addition some of the figure captions need an overhaul for clarity.

RESPONSE TO REVIEWERS' COMMENTS

Reviewer #1 (Remarks to the Author):

The Sadier et al is a very interesting work that explores the diversity of bat dentitions, with a special focus on developmental mechanisms that might regulate and bias their diversity. Bats (here noctilionoids) show remarkable range of dental patterns, and although the phylogenetic and ecological factors underlying this diversity has been studied, the development of the dentitions has achieved much less attentions. This is undoubtedly much to do with the difficulty in obtaining material to study the process of bat tooth development. Here Sadier and co-workers have carried out a major effort to rectify the lack of data on the developmental basis of bat dental diversity. Just providing this basic data is a very valuable contribution worthy of publication. Analytically, the authors frame their results in the context of inhibitory cascade rule of sequential tooth development. They appear to argue that growth of the jaw is a major driver in the regulation of premolar and molar proportions. Below I have a few suggestions to clarify the text and the analyses.

We gratefully thank Jukka Jernvall for his helpful suggestions.

1) Assuming that the premolar proportions follow the inhibitory cascade (IC). IC mode of development assumes sequential development of individualized units with directionally biased effects. This mode of development is rather self evident for the molar teeth which bud off distally. In contrast, in premolars the cascading effects can be assumed to affect principally the deciduous premolars. Hence, why Evans et al 2016 examined deciduous premolars in hominins. Because permanent premolars develop as offshoots from their deciduous predecessors, the IC effects would be very indirect (at best), and I am not sure if there has been an explicit suggestion of permanent premolars following IC (at least I do not remember making one). That said, the analysis provided by the authors is totally valid, as they clearly demonstrate how premolars do not follow the IC (Fig. 2A). What might be in order is to slightly rephrase the text to state that permanent premolar development is not strictly a cascading process similar to molars. Thus, the primary hypothesis for this part of the analysis is not the IC, confirmed by their analyses.

We agree and we have rephrased the text and expanded discussion to integrate this point.

2) Related to the point above, it is currently not clear in the text whether the deciduous dentition of the bats studied are functional. I recall that bat deciduous teeth tend to be reduced in number and size, and the beautiful and informative Fig. 3 seems to imply that this may be the case for at least some of the taxa studied. This would be valuable to point out. Also, if there are data to measure deciduous tooth sizes, this could be used to test the IC on those teeth. But I understand this data may be not available.

Very interesting point. We have added some of the rare papers that investigate tooth renewal in bats to discuss this question. Briefly, bat deciduous premolars are morphologically highly derived with a "hook" shape that allow the pup to attach to the mother's nipple during flight (Spillman, 1927, Reeder et al. 1953 for the fishing bat, Stegeman, 1956 for Myotis (with drawings), Birney and Timm 1975 for vampires, Sigé et al. 1991, Simmons et al. 1994, Sigé et al. 1998 for a fossil of Palaeochiropteryx, or Popa et al. 2016 for another clade of bats in which the "hook" form can be seen during development). Additionally, our personal observations of bats caught in the wild carrying pup have allowed us to observe this behavior (however, we always release mothers with infants). Bat deciduous dentition is thus functional but functionally and morphological derived. However, the respective function of dP3 and dP4 are still unclear as

both can potentially have a role in attachment during flight - are they both functional? Redundant? Further studies in the wild on a large number of individuals might help clarify these questions. Additionally, we have looked for such specimens (juveniles and new born) in the collections of the AMNH (New York, USA) and the Burke museum (Seattle, USA) but failed to find enough data for these analyses to be relevant. Such specimens will be captured in the wild for future studies.

3) The Gierer-Meinhardt modeling is very well explained in the supplement. One detail that should be corrected in the main text (Page 2-3) is that the original IC model is not a dynamic reaction/diffusion model, but rather a rule assuming a simple activator/inhibitor balance in determining tooth proportions (this does not exclude the possibility of a Turing like process). Considering the modeling in the Sadier et al, the domains signifying tooth loci presumably denote to the placodes (initiation knots), although they could be considered to represent forming primary enamel knots too. In either case, the locations ('cells') representing signaling centers appear to move (change locations) in some of the model runs (e.g. SI Fig. 9). This is to be expected in dynamic systems without terminal differentiation. If this were the case in real jaws, the implication would be that the identity of placodal cells would shift before becoming fixed. Currently I did not notice mentioning of the dynamic nature of the modeling, and I leave it to the authors to decide whether something to point out.

We have rephrased our discussion of reaction/diffusion and the IC model in the main text. Regarding the movement of the patterning structures, this is now briefly discussed in the supplementary text as due to the timescales of Turing instability and pattern formation being much faster than the timescales of growth. This leads to effective movement of emergent structures, which in principle could be corrected with more realistic models of cell differentiation or tissue restructuring, as suggested.

4) Perhaps the most challenging part of the paper for a more general readership is the Morphogroups presented (3P3MR, 3P3ML...). These are very difficult to follow and the codes themselves are rather cryptic. It would make the text more readable if at least the main groups would be described in more understandable terms (superficially the dentitions seem to differ how laterally compressed they are). Also, currently the point colors in Fig. 2 seem to refer to the Morphogroups, although this is not stated. Interestingly, some groups (3P3ML?) seem to follow the IC prediction quite closely in Fig. 2B, as also many of the green group taxa where the M3 is highly reduced are close to the IC line.

We have changed the morphogroups names and replaced them with more generic names for clarity. We called the different groups "long jaw", "regular jaw" "intermediate jaw" and "short jaw" to be consistent with the message of the paper. We have added sentences to comment on the width of the teeth of the different groups (which indeed differ in how laterally compressed they are). We also added some comments about this observation that some groups seem to follow the DIC more closely than others and discussed this also in the discussion. We kept some of the "dental formula" terminology in some of the Extended Figures since the dental formula is important for statistical analyses.

5) There are some minor issues in text that could benefit from rewording. The use of superlatives such as astonishing, tremendous, groundbreaking, and outstanding is at least to this referee a bit distracting. Additionally, "key event" might be better to be "key innovation" (Introduction, first paragraph).

We have reworded these superlatives and changed key event by key innovation.

Jukka Jernvall

Reviewer #2 (Remarks to the Author):

Overall this is an intriguing paper. The authors have demonstrated an independent cascade pattern for the premolar region relative to the molar region in noctilionid bats, and developed a model based on reaction diffusion gradients in which jaw elongation can modulate the appearance of tooth loci. The data analysis and methodology seem to be sound and original. However, in my opinion, the conclusions are somewhat oversold as explaining more about the evolution of heterodonty and patterning in these regions. The authors admit near the end of the article, this is a complex problem and a number of factors are likely to operate, yet the bulk of the MS states that this model solves everything and we can ignore homeobox patterning completely, even though the model is operating much later in development than homeobox patterning. Further, it seems to take as given that the ICM is the ancestral pattern in mammals, while ignoring that it doesn't apply to many groups, and this warrants further discussion in the MS -- especially since their model only explains the patterns in 2 of the 4 morphogroups they establish, so only about half of the observed range of variation. In addition some of the figure captions need an overhaul for clarity.

We thank reviewer two for their helpful suggestions to clarify the manuscript for a broader readership and provided more details about the developmental mechanisms in play. Below are the detailed responses:

However, in my opinion, the conclusions are somewhat oversold as explaining more about the evolution of heterodonty and patterning in these regions.

We have added more text in the introduction to bring more context about the role of Turing mechanisms – and developmental cascades such as the ICM - especially since it has been often mis-used to infer developmental mechanisms from solely morphological observations (which is not the case in the initial Kavanagh et al. 2007 paper). In this paper, we are studying the evolution of the Turing mechanisms that pattern the jaw (which the DIC model is a particular case of), and how these mechanisms evolve to produce the variation of teeth seen in mammals both from a morphological AND developmental point of view (which has been rarely done since the publication of Kavanagh et al. 2007 as the ICM model has been often testing only through measurement on adult dentitions).

The authors admit near the end of the article, this is a complex problem and a number of factors are likely to operate, yet the bulk of the MS states that this model solves everything and we can ignore homeobox patterning completely, even though the model is operating much later in development than homeobox patterning.

We have amended the introduction for clarity: our intention was never to give the impression that the model “solves everything”, and we acknowledged the complexity of the system, as well as the existence of the homeobox code (see the reference to Wakamatsu et al. 2019). We have also added more references here “As tooth proportions are controlled by the underlying developmental mechanisms that regulate their formation” that all point out that the patterning and later events are important to regulate tooth size. However, because developmental systems, patterning and morphogenesis are so complex, it is nearly impossible to study all the

mechanisms at the same time (an iconic example is digit patterning, see Raspopovis et al. 2014, in which digit patterning is explained by a self-organizing “Turing-like” mechanism but in which earlier positional effects are in play see Delgado and Torres 2016 or Tickle and Towers 2017). Here, we use the predictions of the DIC (i.e. do premolars and molars follow the cascade?) as a starting point to study the evolution of the underlying developmental mechanisms that pattern the different tooth classes. We agree with the reviewer that it might not be obvious for readers less familiar with tooth developmental biology and have added some context to highlight how complex the patterning of the jaw is. While all teeth (and indeed other ectodermal appendages) undoubtedly share common mechanisms for their formation (see below), our results suggest that the two classes are patterned by different developmental cascades (that are largely not interacting) suggesting that they develop independently (something that can be coupled or decoupled from the hox code pre-patterning). To our knowledge, this has never been reported. However, we agree with the reviewer that we are not investigating earlier steps, in particular the pre-patterning of the different regions of the jaw that is hypothesized to influence the determination of tooth classes (prior the appearance of tooth buds and the patterning of the teeth). As a result, the homeobox code is not ignored but is beyond the scope of this paper which studies later events (from bud to bell) that also participate to the acquisition of tooth identity. We have modified the discussion to highlight this and have modified the introduction to explain the focus on the tooth patterning events (i.e. the successive emergence of signaling centers).

Further, it seems to take as given that the ICM is the ancestral pattern in mammals,

We have edited the manuscript for clarity since this is not what we stated nor what we meant (see citations in the introduction, we agree with the reviewer that this is likely not the case), but we understand this comment as it is often assumed that the ICM is ancestral. The ICM has been established in rodents for molars both from a morphological and developmental point of view and is a particular case of the general mode of ectodermal appendages patterning with a directional growth. While not necessarily ancestral, it remains a good starting point to study how developmental mechanisms differ (in the sense of developmental biology) between species from the only one that has been well studied from a molecular and developmental point of view (in mice).

while ignoring that it doesn't apply to many groups,

We are sorry the reviewer has this impression We did not ignore this (see the citations in the introduction, including Polly et al. 2007) but we recognized that we should have been more explicit. We have rewritten sentence to clarify this, in particular since this is an important message of our paper. It is even the starting point to study how these cascades evolve between species in order to produce the diversity of shape that are seen in mammals. We agree with the reviewer that the term “cascade” is often linked to the original ICM described in rodents. We have expanded this part to make this statement clearer and explained that the ICM is just a particular case of a Turing pattern (which is well explained in Zimm et al. 2022).

and this warrants further discussion in the MS -- especially since their model only explains the patterns in 2 of the 4 morphogroups they establish, so only about half of the observed range of variation.

We are puzzled by this sentence. Our model explains the variation seen in all the morphogroups (see Figure 5 and our supplementary material), by incorporating growth into the Turing mechanisms in growing domains. However, it is true though that the original ICM equations only explain the variation seen in 2 morphogroups (for the molars) and that is the reason why we

incorporate growth rate variation in our model. We have clarified the related sentences to avoid any confusion between our model and the original ICM.

In addition, some of the figure captions need an overhaul for clarity.

We agree and have clarified the figure captions for clarity.

What precisely is meant by this? Couldn't figure out.

We meant that we can study these species from a developmental point of view (it is not really possible to draw strong conclusions solely from morphological measures in the adult dentition) to study the formation of tooth buds and/or perform molecular experiments. We have expanded the sentence for clarity.

Aren't cell division rates a form of heterochrony?

Heterochrony is a change in the rate or timing of development relative to the ancestor (see McNamara, 2012). Often, two types of heterochronies are distinguished: peramorphosis and paedomorphosis. Cell division rate is one of the mechanisms by which heterochrony can be achieved but not a "form" of heterochrony.

This is arguable - they didn't uncover it, they proposed a model - and it doesn't fit several groups of mammals as proposed. See

We have edited the opening sentence for clarity and have reorganize the paragraph to make this more obvious. Indeed, the authors uncovered a particular Turing pattern, whose dynamic acts as a cascade (as explained in Zimm et al. 2022) that is likely to explain the successive development of molars in most mammals (given that tooth buds appear in a proximo-distal manner as the dental lamina grows). However, we agree with the reviewer that the proportions calculated from the mathematical model vary between clades, implying the existence of other developmental mechanisms which is one of the main messages of our paper.

Why no citation to Turing's work here?

We agree and have added the citation.

Growth of what is being referred to here? Presumably elongation of the jaw?

Growth is used here in its general sense (as used in developmental biology), i.e. growth of a field, domain, tissue. We have added "tissue" for clarity.

You propose three alternate models in the figure 2D so I'm confused with this sentence - is the direction not important?

We have edited the sentence from clarity. Three models are possible to explain the development of postcanine teeth: 1) one initiation from the canine (as hypothesized based on mice whose development is very derived) with a proximo-distal apparition of the buds; 2) one initiation that develop in both directions; 3) two independent initiations that develop in opposite directions. If premolars and molars are initiated as in 1), we expect the triplets to follow the DIC (or closely). If premolars and molars are initiated as in 2) at M1, we expect the M1-M2-M3 (proximo-distal direction) or the P3-P4-M1 (disto-proximal direction) proportions to be linked. See Evans et al. 2012 for more details.

There is a lot of work done on the pre-patterning of the tooth buds and how dental identities may be established prior to the bud stage of development - see Zimm et al. 2022 "Turing's turtles all the way down: A conserved role of EDAR in the carapacial ridge suggests a deep homology of prepatterns across ectodermal appendages". This casts some doubt on the interpretation here.

We have clarified this sentence as we agree that the term result is confusing (see below). While there is likely a deep homology of ectodermal appendages (see Mustonen et al. 2004; Dhouailly, 2009; Biggs and Mikkola, 2014; Sadier et al. 2014; Zimm et al. 2022), which is exemplified by the mechanisms behind their development, including reaction/diffusions mechanisms of patterning, it is not exclusive with their development/patterning being independent from each other's (i.e. hair development is independent from the development of teeth or mammary glands spatio-temporally and part of the developmental genes, developmental mechanisms and core GRN modules involved). We are confused by the reference regarding tooth class development: the tooth work cited in Zimm et al. 2022 on the pre-patterning refers to the determination of the odontogenic mesenchyme that will then give rise to the dental lamina, and even later to the dental placodes (Peters and Balling, 1999). Further, this study refers to the patterning of the premolars and molar buds by cascade type mechanisms, citing our previous work (Sadier et al. 2019) as well as the original ICM work (Kavanagh et al. 2007) which focused on molar patterning. In addition, Zimm et al. 2022 describe the role of Edar in modulating the size of the teeth and the placodes citing various work including ours. Regarding teeth, this work focusses on the patterning of the tooth buds (and eventually the pre-patterning of the Turing mechanisms with a large field expressing Edar) but not the pre-patterning of the jaw and the hox code (see Wakamatsu et al. 2019 for the most recent paper) that happens prior the budding of the dental lamina.

The claim that regional identity receives contributions from two different cascades post-bud stage seems well supported by these data - but I think its a bit of a stretch to say regional identity is entirely determined by two completely independent cascades - later on you are more conservative and note other contributors - feels like this is being over-sold here.

We have changed the term "result" by "are patterned by" which is more accurate. Indeed, our results show that premolars and molars develop independently (i.e. from placode induction to bell stage which is strictly tooth development see Tucker and Sharpe, 2003 and Caton and Tucker, 2009 or Thesleff et al. 2003) and not that "the regional identity is entirely determined by two independent cascades" (indeed, the regional identity of the jaw is established earlier, prior to the induction of tooth signaling centers, see Caton and Tucker. 2009 for a review). We agree with the reviewer that different signals are likely to be integrated for the determination of tooth identity. How the presumptive territories that are determined prior placode induction (pre-patterning of the jaw) by the hox code is beyond the scope of this paper and happened prior placode induction (see Caton and Tucker, 2009). It will be particularly interesting to try to link the hox code with the different cascades in future studies. We do not understand the "post-bud stage" statement, the DIC control the patterning (i.e. the apparition of the placode) pre-bud, not post-bud (see Kavanagh et al. 2007, Prochazka et al. 2010, Sadier et al. 2019 and Zimm et al. 2022).

Again, a bit overstated given the findings and other work.

We have edited the sentence for clarity. Indeed, in model species, it has been shown that tissue growth is important for the correct spacing of serial organs (see Mou et al. 2006, Kondo et al. 2010, Fetaher 2019, Zimm et al. 2022, Economou et al. 2012). However, how variation in growth rate influences the diversification of tooth morphology and facilitates variation in an adaptive context has never been demonstrated. Here, we see a strong correlation between jaw

size and tooth number, and, given what is known about the patterning of ectodermal appendages, we postulate that this mechanism might have driven the rapid evolution of tooth number and size in this clade. Our conclusions are supported by the convergent phenotype in multiple species, our modeling simulations and the molecular data in multiple species.

Given this model only explains half of the morphogroups you produced, I am not convinced that this is warranted.

We have edited for clarity since there might have been a misunderstanding between the original ICM (which indeed explains half of the morphogroups) and our model that combine Turing patterns in a growing domain (ICM is a particular case) that include differential growth which explains the variability observed in the morphogroups. These findings confirm the results of Zimm et al. 2022 and, more largely, other ectodermal appendages.

Again, I'd argue this oversells the results.

We have edited the sentence. To our knowledge, the most recent papers regarding the origin of the tooth classes also state that the mechanisms behind tooth class determination are poorly understood (Wakamatsu et al. 2019 for a revised homeobox code and Woodruff et al. 2022 which has been published after our submission).

Again, this supports that regional identity is at least partially influenced by independent mechanisms ONCE THE BUDS HAVE FORMED, but it completely discounts any earlier patterning.

We have edited the sentence for clarity (and replaced mechanisms by activation/inhibition mechanisms after the initial patterning of the jaw, which is what is discussed in this paragraph). The study of the pre-patterning of the jaw and regional identity is beyond the scope of this paper. Here, we study the patterning of premolars and molars (i.e. the successive apparition of premolars and molars) and not the regionalization of jaw (i.e. regional identity). These events pattern the tooth placodes (pre-bud) and we are able to visualize the epithelial thickening (pre-bud) see Kavanagh et al. 2007, Tucker and Sharpe 2004, for the successive order of events. In addition, other mechanisms are likely at play, see Moustakas et al 2014 and Woodruff et al. 2022 for gene expression variation between tooth classes. We also added a sentence at the end of the paragraph to make clear that the hox code pre-patterning of the jaw likely influence the later patterning of tooth buds.

I fail to see how this growth mechanism supersedes earlier patterning completely. Far more likely that both mechanisms contribute to tooth class identity as neither one seems sufficient on their own to establish this. Also this clause needs a reference.

We are puzzled by this comment: variation in growth rate influence the patterning of premolars and molars (i.e. the successive apparition and the overall tooth size) but indeed, do not supersedes the regionalization of the jaw (as we explain here: "The work we present here, which suggests that premolars and molars develop independently by two different cascades, supports the hypothesis that the development of at least the premolar and molar tooth classes are largely independent and uncovers a mechanism that could act in tandem with the homeobox code and other processes to modulate that development"). As stated, both contribute likely to the determination of tooth identity but, up to now, the existence of distinct cascades has not been described. The references are all listed in the following sentences. We have edited for clarity and to extent the scope of these results.

Again - this is only post-bud and presupposes the ICM is the primary mechanism used to determine dental morphology. I do not think this is an entirely warranted assumption.

We have edited for clarity as we do not state nor even hypothesize that the ICM is the primary mechanisms to determine dental morphology (Turing mechanisms contribute to this, as for other ectodermal appendages). Although Turing patterns induce placode formation in ectodermal appendages and teeth (see Mou et al. 2006, Kavanagh et al. 2007, Sadier et al. 2019, Zimm et al. 2022), which is pre-bud.

It's pretty well established that spacing is important in pre patterning ectodermal appendages - see EDAR turtle paper again.

This has been shown to be important in foundational papers which are cited such as Kavanagh et al. 2007; Polly et al. 2007; Young et al. 2015, Kondo et al. 2010, Economou et al. 2012 and 2020 and also mentioned by Zimm et al. 2022. Since growth is important for Turing patterns, we modeled Turing patterns in a growing domain and hypothesized that the variation in growth rate is also able to generate a variation of serial organs number and overall size. Indeed, in Zimm et al. 2022, it is postulated that "Thus, we find that the development of ectodermal appendages represents a nested sequence of more or less independent, deployable patterning modules, most of which involve reaction–diffusion systems and growth." and idea that we also develop in our review (Sadier et al. 2020). Because of this, we postulate that our model could explain the evolution of the patterning of other ectodermal appendages in multiple species which was not been demonstrated (i.e. variation in growth). Our model is novel in the sense that it provides a new testable framework to model to predict and explain the variation of serial organs number and size. To our knowledge, this has not been demonstrated before (i.e. how these mechanisms generate the variation seen in nature).

Are there any potential issues with basing the staging on this one species?

We have added more explanations about the issue of staging and the use of the Carnegie Stage (CS) system. Regarding teeth, we have verified that the first premolar and molar appear at the same CS and have followed their development for each developmental stage as previously (Camacho et al. 2020)

REVIEWERS' COMMENTS

Reviewer #1 (Remarks to the Author):

I have now reread the entire manuscript by Sadier et al (incidentally, the rebuttal letter lacked line numbers to revisions which slowed things a bit). The authors have addressed the comments reasonably and I find the text improved and easy to follow. The data is impressive, and the conclusions seem well measured. The authors argue for the role of the jaw in the regulation of tooth number and size, which together with the previous proposals of jaw facilitating tuning of cusp patterns, raise the general question of the autonomy and integration of organ development. The work will undoubtedly stimulate new studies to test different points of views.

There are some figures where text is partially hidden (Fig 1b, 6a) probably by the a,b,c-lettering.

Reviewer #2 (Remarks to the Author):

The MS reports work on noctilionoid bats, using a combination of mechanistic modelling and work with imaging of the dental lamina. This clade of bats shows a high disparity of jaw lengths and dental morphology, and provides a good alternative to mouse as a developmental model for mammalian dental evolution (having all of the tooth classes represented is critical for this work). The work improves upon the existing modelling for diffusion gradients in the size and spacing of teeth by adding a growth component to the models. This allows better prediction across this clade, and identifies an important component in the evolution of regionalization of the dentition. The work as described in this version of the MS supports the conclusions they present, the methodology is sound and meets expected standards, and there is sufficient detail to reproduce the work.

This is a novel and important contribution of wide interest in developmental biology, evo-devo, and more broadly in anatomy and paleontology.

The edits have significantly improved the clarity of the MS and made the main points accessible to a much broader audience. I enjoyed this version very much and look forward to citing it.

A few more line edits:

Introduction:

L61 – Maybe some re-phrasing here? Such as “variation in heterochronies during development, some of which involve changes in cell division rate”?

L62 – deciduous is the common English usage (you use decidual in several places, I usually see that phrasing with reference to placenta rather than teeth)

L 71 – successive emergence of molar

L74 – cascade type might be the

L82-83 – Tooth signalling centers – ambiguous, could apply to a variety of things, what stage of development do you mean here or is this meant as catchall?

L 92 – 95 – This sentence is very awkward – is the comma after growth perhaps better placed after Turing processes? “by perturbing the underlying Turing processes, growth modulates the number and size of the different classes of the teeth of noctilionoid bats during their adaptation to various dietary niches”

L 96-98 – Saying evolution uses things implies agency. Alternative phrasing that states “evolution of ectodermal appendages may occur through changes in similar patterns...” or “Turing patterning is modulated by evolution” would be less problematic.

Results:

L015 – formatting error with “. 28.”

L122 - premolars and molars proportions

L143 – ?Use “correlated with” instead of “tied”, the heading implies causation before you present your argument

L154 – duplicate short jaw / short jaw

L186 – extraneous period at end

Discussion:

L256 - premolars and molars placodes

L261 - prior to placode

L264 - tooth classes identity

L265 - such as the lizard genus Pogona

L279 - This is great! You really clearly emphasize that the initiation and morphogenesis of the premolars and molars are independent - this wasn't completely clear before, but it is now.

L 282, 287 - Would avoid the use of the term colonization where it isn't necessary. Synonyms like invasion or exploring would work.

L282 - 298 - Really like this paragraph. You very clearly state how growth rate could be one of the mechanisms used to escape the ICM. This is amazing work!

L314-316 - Use of modulate three times in a single sentence

L331 - I am not sure developmental biases are the right word, this has a bit of a negative connotation. Maybe rephrasing to "among many developmental inputs/levels" ? Bias implies a constraint rather than opening of pathways for morphological divergence which is your main argument.

REPOSE TO REVIEWERS' COMMENTS

Reviewer #1 (Remarks to the Author):

I have now reread the entire manuscript by Sadier et al (incidentally, the rebuttal letter lacked line numbers to revisions which slowed things a bit). The authors have addressed the comments reasonably and I find the text improved and easy to follow. The data is impressive, and the conclusions seem well measured. The authors argue for the role of the jaw in the regulation of tooth number and size, which together with the previous proposals of jaw facilitating tuning of cusp patterns, raise the general question of the autonomy and integration of organ development. The work will undoubtedly stimulate new studies to test different points of views.

We thank Reviewer 1 for his helpful suggestions.

There are some figures where text is partially hidden (Fig 1b, 6a) probably by the a,b,c-lettering.

This has been updated to correct formatting issues.

Reviewer #2 (Remarks to the Author):

The MS reports work on noctilinoid bats, using a combination of mechanistic modelling and work with imaging of the dental lamina. This clade of bats shows a high disparity of jaw lengths and dental morphology, and provides a good alternative to mouse as a developmental model for mammalian dental evolution (having all of the tooth classes represented is critical for this work).

The work improves upon the existing modelling for diffusion gradients in the size and spacing of teeth by adding a growth component to the models. This allows better prediction across this clade, and identifies an important component in the evolution of regionalization of the dentition.

The work as described in this version of the MS supports the conclusions they present, the methodology is sound and meets expected standards, and there is sufficient detail to reproduce the work.

This is a novel and important contribution of wide interest in developmental biology, evo-devo, and more broadly in anatomy and paleontology.

The edits have significantly improved the clarity of the MS and made the main points accessible to a much broader audience. I enjoyed this version very much and look forward to citing it.

We thank reviewer 2 for their helpful feedback on the manuscript.

A few more line edits:

Introduction:

L61 –Maybe some re-phrasing here? Such as “variation in heterochronies during development, some of which involve changes in cell division rate”?

We have rephrased in accordance to the paper cited.

L62 – deciduous is the common English usage (you use decidual in several places, I usually see that phrasing with reference to placenta rather than teeth)

Corrected

L 71 – successive emergence of molar

Corrected

L74 – cascade type might be the

Corrected

L82-83 – Tooth signalling centers – ambiguous, could apply to a variety of things, what stage of development do you mean here or is this meant as catchall?

We have added “primary”.

L 92 – 95 – This sentence is very awkward – is the comma after growth perhaps better placed after Turing processes? “by perturbing the underlying Turing processes, growth modulates the number and size of the different classes of the teeth of noctilionoid bats during their adaptation to various dietary niches”

Corrected

L 96-98 – Saying evolution uses things implies agency. Alternative phrasing that states “evolution of ectodermal appendages may occur through changes in similar patterns...” or “Turing patterning is modulated by evolution” to would be less problematic.

We agree and have edited

Results:

L015 – formatting error with “. 28.”

L122 - premolars and molars proportions

Corrected

L143 – ?Use “correlated with” instead of “tied”, the heading implies causation before you present your argument

Corrected

L154 – duplicate short jaw / short jaw

Corrected

L186 – extraneous period at end

Corrected

Discussion:

L256 - premolars and molars placodes

Corrected

L261 - prior to placode

Corrected

L264 – tooth classes identity

Corrected

L265 - such as the lizard genus Pogona

Corrected

L279 – This is great! You really clearly emphasize that the initiation and morphogenesis of the premolars and molars are independent – this wasn't completely clear before, but it is now.

Thanks!

L 282, 287 – Would avoid the use of the term colonization where it isn't necessary. Synonyms like invasion or exploring would work.

We have changed “colonization of the morphospace” by “exploration”.

L282 – 298 – Really like this paragraph. You very clearly state how growth rate could be one of the mechanisms used to escape the ICM. This is amazing work!

Thank you very much!!

L314-316 – Use of modulate three times in a single sentence

Corrected.

L331 – I am not sure developmental biases are the right word, this has a bit of a negative connotation. Maybe rephrasing to “among many developmental inputs/levels” ? Bias implies a constraint rather than opening of pathways for morphological divergence which is your main argument.

Edited, indeed, the ICM is considered as a developmental constraint/bias but not necessarily the other parameters.